# Evaluating the Effect of Image Enhancement on Diagnostic Reliability in Dry Eye Disease Using a Portable Imaging Device

**DOI:** 10.3390/diagnostics14222552

**Published:** 2024-11-14

**Authors:** Takahiro Mizukami, Shinri Sato, Kazuki Asai, Takanobu Inoue, Eisuke Shimizu, Jun Shimazaki, Yoshikazu Shimomura

**Affiliations:** 1Department of Ophthalmology, Fuchu Hospital, Osaka 5940076, Japan; y_shimomura@seichokai.or.jp; 2Yokohama Keiai Eye Clinic, Yokohama 2400065, Japan; shinri.sato259@keio.jp (S.S.); asai@keio.jp (K.A.); ophthalmolog1st.acek39@keio.jp (E.S.); 3OUI Inc., Tokyo 1070062, Japan; takanobu.inoue.ds@gmail.com; 4Akasaka Shimazaki Eye Clinic, Tokyo 1070052, Japan; meishano1@gmail.com

**Keywords:** dry eye disease, smart eye camera, tear meniscus height, conjunctivochalasis, interobserver reliability, image enhancement

## Abstract

Background: This study aimed to evaluate the impact of image enhancement techniques on the interobserver reliability of tear break-up time (TBUT), tear meniscus height (TMH), corneal fluorescein staining (CFS) scoring, and conjunctivochalasis detection using the Smart Eye Camera (SEC), a portable device for anterior segment examination. Methods: A retrospective analysis was conducted on video recordings captured by the SEC from 46 patients with dry eye disease (DED). Separate sets of images were created for each level of enhancement: unenhanced (G0), mildly enhanced (G3), and strongly enhanced (G7). These sets were not intermixed, ensuring that each enhancement level was assessed independently. Three observers—two DED specialists and one general ophthalmologist—assessed TBUT, TMH, CFS scores, and conjunctivochalasis. Interobserver reliability was evaluated using intraclass correlation coefficients (ICCs) for each image set. Results: Interobserver reliability for CFS scores significantly improved with G3, yielding an ICC of 0.8413. In contrast, G7 improved reliability for TBUT measurements (ICC = 0.7381), but led to a notable decrease in reliability for both CFS scoring (ICC = 0.2259) and conjunctivochalasis detection (ICC = 0.0786). Furthermore, the assessment of TMH demonstrated a progressive decline in accuracy with increasing levels of image enhancement. Conclusions: Image enhancement using the SEC improved the diagnostic consistency of dry eye specialists and general ophthalmologists, especially for TBUT and CFS assessments. However, excessive enhancement may obscure key diagnostic features, indicating the need for careful optimization of image processing techniques depending on the diagnostic focus.

## 1. Introduction

Dry eye disease (DED) is a multifactorial disease characterized by unstable tear film causing a variety of symptoms and/or visual impairment, potentially accompanied by ocular surface damage [1]. Contributing factors include age, female gender, the use of certain medications, underlying medical conditions, digital screen exposure, and the use of cosmetics, contact lenses, or eye drops [2,3]. Environmental influences such as hot, dry weather and outdoor occupations may also play a role [4]. DED can have a significant impact on quality of life, visual performance, and general health [2]. The economic costs are considerable, with healthcare expenditures for DED patients in Japan estimated at $323 annually and productivity losses ranging from $741 to $6160 per year [5]. Objective methods for evaluating DED include the tear break-up time (TBUT) test, which measures tear film stability, and fluorescein staining used to assess corneal and conjunctival damage [6]. Previous reports indicate that software-detected non-invasive break-up time (NIBUT) exhibits greater sensitivity in detecting tear break-ups and can identify specific disruptions in the tear film that may be overlooked by traditional TBUT methods [7]. Subjective symptoms can be evaluated using the Ocular Surface Disease Index (OSDI) questionnaire [8,9]. The Asia Dry Eye Society highlights the essential role of TBUT assessment, in which fluorescein is employed [1].

To evaluate TBUT and other anterior segment abnormalities caused by DED, examination using a slit-lamp biomicroscope is typically required [1]. However, conventional slit lamps are costly and require trained professionals for operation, limiting their use to hospitals or specialized eye clinics. Consequently, they are impractical for deployment in remote or underserved regions. Portable slit lamps offer a more accessible alternative due to their portability; however, their high cost and inability to capture images for documentation remain significant limitations [10]. In response to this limitation, the Smart Eye Camera (SEC), a novel portable device designed for ophthalmological examinations, offers a practical solution [11]. This innovative technology converts a standard smartphone camera into a tool capable of performing basic assessments of the anterior segment, including the ocular surface. The SEC is as reliable as the conventional non-portable slit-lamp microscope for evaluating nuclear cataracts, anterior chamber depth evaluation, TBUT, corneal fluorescein staining, and tear meniscus height (TMH) [11,12].

TBUT reliability can be influenced by measurement techniques, observer experience, and assessment methods [13]. The high reliability of interobserver assessment indicates that DED diagnosis using video taken with SEC is reliable [14]. The corneal fluorescein staining (CFS) score is commonly employed in the assessment of ocular surface damage [15]. Although TMH is a reliable metric for diagnosing DED, the measurement can vary considerably depending on the instrument used; the SEC demonstrated sufficient validity and reliability in assessing TMH, showing concordance with conventional slit lamp examination [11]. As previously mentioned, the use of the SEC for diagnosing DED has shown results comparable to conventional slit-lamp examination. However, a future challenge will be enabling general ophthalmologists, who do not specialize in the cornea, to achieve the same diagnostic accuracy as dry eye specialists. One potential solution is the application of image processing techniques to enhance the clarity of key visual features. In fact, in other fields, various applications of image processing have been reported, as exemplified by the following cases. Color enhancement and achromatization using digital-assisted vitrectomy offer advantages to enhance the visibility of indocyanine green-stained internal limiting membrane for peeling [16]. Texture and color enhancement imaging optimizes the visualization of structure, color tone, and brightness and improves the visibility of gastric tumors allowing their early detection when performing esophagogastroscopy [17,18]. By applying appropriate image processing techniques to videos captured by the SEC, it may become possible to enhance the visibility of desired features, thereby facilitating a more accurate assessment of the ocular surface condition.

The goal of the present study was to investigate whether applying various image enhancement techniques to SEC video influenced the agreement rates between dry eye specialists and a general ophthalmologist, specifically in the measurement of TBUT, the assessment of TMH, and the evaluation of CFS score, as well as the diagnosis of conjunctivochalasis.

## 2. Materials and Methods

### 2.1. Study Population

This retrospective, monocentric study was carried out at the Yokohama Keiai Eye Clinic, Japan, between July 2020 and December 2021. This research project received approval from the Institutional Ethics Review Board of the Minamiaoyama Eye Clinic, Tokyo, Japan (IRB No. 202101) and the Ethics Review Committee of the Kobanawa Medical Corporation (Committee No. 21000056), ensuring compliance with ethical standards outlined in the Declaration of Helsinki. Written informed consent was not obtained because this study was retrospective. Patients were recruited from among those attending routine visits at the Yokohama Keiai Eye Clinic. The Japanese version of the OSDI questionnaire was used to measure subjective indicators of DED [19].

### 2.2. Inclusion and Exclusion Criteria

To be eligible for this study, subjects had to meet the inclusion criteria, i.e., exhibit no signs of active ocular inflammation, and have no history of ocular diseases or surgeries, except cataract surgery performed at least six months before this study. Conversely, the exclusion criteria were as follows: the use of ocular hypotensive drops, wearing contact lenses, and a confirmed diagnosis of any chronic ocular disease. This careful selection of participants was designed to ensure a homogeneous study population and reliable results [20].

### 2.3. Smart Eye Camera

The SEC is a portable device that functions as a slit lamp. This device has received medical device approval in Japan (registration number: 13B2X10198030101), Europe (bearing the “CE” mark), Kenya, Vietnam, Cambodia, and Indonesia. The SEC’s main body is constructed from polyamide 12 using 3D printing technology, and it is designed to be mounted on an iPhone 8 (Apple Inc., Cupertino, CA, USA). The video capture resolution is 1080p at 30 frames per second, yielding an output of 2.1 megapixels (2,073,600 pixels) per file [11]. The SEC is equipped with interchangeable lenses, including a convex macro lens (focal length = 20 mm, magnification = ×20) for diffuse illumination, a slit lamp converter with a fixed slit width of 1 mm, and a blue filter. Slit illumination is achieved using a cylindrical lens positioned over the smartphone camera at a fixed angle of 40° [11]. The blue filter, composed of acrylic resin, emits light at a wavelength of 488 nm [21]. The lenses can be manually switched by sliding them along the base structure’s designated guides, ensuring precise and stable positioning during examination. This imaging system is operated through a user-friendly dedicated application, enabling seamless access to the smartphone’s camera for video capture. The app also provides secure storage options, both online and offline. Additionally, a teleconsultation feature is integrated into the app, facilitating remote ophthalmological evaluations. This feature allows users to engage in virtual consultations with healthcare professionals, offering a practical solution for both patients and clinicians in ophthalmology [22].

### 2.4. Capturing Anterior Segment Videos

In this study, each patient received 2 μL of a 0.5% diluted sodium fluorescein solution (Fluorescite™; Novartis Japan Ltd., Tokyo, Japan), which was carefully instilled into the inferior conjunctival sac using a micropipette to ensure precise and uniform dosing across all participants [21]. Following instillation, videos were recorded using the SEC exactly 3 min after administration. This interval was critical to ensure adequate distribution and interaction of the fluorescein with the ocular surface. Videographic sessions were performed in a dark room to limit bias due to high illumination. Each patient was subjected to SEC videography with a blue filter at around 4 cm distance from the corneal apex, as the convex lens in front of the camera was designed to be in the best focus by 4 cm.

### 2.5. Image Enhancement

Three versions of videos were prepared for evaluation using the following methods.

Parameter Adjustment: In this study, the green (G) channel of the RGB color model was selectively enhanced by multiplying its intensity by a factor of *n*. This adjustment was made to emphasize the green component without causing excessive brightness or saturation that could obscure important visual details in the video. To ensure the quality of the enhancement, a manual inspection of each processed video was performed to confirm that the image remained clear and that excessive brightness or “blown-out” areas did not occur.

Software and Tools: The analysis and processing of videos were conducted using Python (version 3.10.15) programming language, specifically leveraging the MoviePy (version 1.0.3) and OpenCV (version 4.9.0.80) libraries. These tools were utilized for frame extraction, RGB manipulation, and recombination into video files. OpenCV was used to read and modify pixel values, while MoviePy was employed for handling video input and output operations, including the application of RGB adjustments to each frame and re-encoding the video.

Procedure and Code Implementation: This process involved iterating through each frame of the input video files, adjusting RGB color channels, and re-saving processed videos. The core function, process_frame, manipulated pixel values by scaling red, green, and blue channels according to predefined ratios. The green channel was adjusted by multiplying its value by *n*, while the red and blue channels were left unchanged or scaled by different ratios depending on the experiment.

Video Processing: Videos were processed frame-by-frame, with each frame’s green channel multiplied by the specified factor (e.g., 7×), while leaving the other channels unchanged. After processing, frames were recombined into a video file, which was saved in MP4 format using the H.264 codec. During this procedure, the resolutions of these videos were maintained at their original values to ensure consistency in visual quality.

Manual Review: Following video processing, each output file was visually inspected to verify that the image quality was maintained, ensuring that excessive brightness (“blown-out” highlights) did not occur due to the green channel enhancement. This step was crucial to confirm that adjusted videos were suitable for further analysis or presentation.

### 2.6. Image Analysis

In this study, TBUT, TMH, CFS scores, and the presence of conjunctivochalasis, as assessed from video recordings captured by the SEC, were evaluated by three independent observers (E.S. and S.S., dry eye specialists, and T.M., general ophthalmologist). Observers were classified as dry eye specialists if they had over five years of experience in a dedicated dry eye clinic. Three versions of each video were prepared for evaluation by three independent observers based on the following criteria. The original video without image enhancement was designated as G0, the video with mild enhancement using the method mentioned above was labeled G3, and the video with stronger enhancement was labeled G7. Representative images for each condition (G0, G3, and G7) are shown in Figure 1.

TBUT was measured three times per eye for each patient, and the average TBUT by each observer was used for analysis [23]. TBUT was evaluated at G0, G3, and G7. Figure 2 illustrates the appearance of tear film break-up under the conditions of G0, G3, and G7.

TMH was evaluated for each eye on a three-point scale by the observers (high = 2, middle = 1, and low = 0) [24]. TMH was evaluated at G0, G3, and G7. Figure 3 illustrates TMH under the conditions of G0, G3, and G7.

CFS scores were graded using the Ocular Surface Disease Index proposed by Ogawa et al., dividing the cornea into upper, middle, and lower regions [25]. Each region was scored on a scale of 0 to 3, with a maximum score of 9 (Grade 0 = no staining, Grade 1 = minimal staining, Grade 2 = mild/moderate staining, and Grade 3 = severe staining). CFS scores were evaluated at G0, G3, and G7. Figure 4 illustrates corneal fluorescein staining under the conditions of G0, G3, and G7.

For conjunctivochalasis, a score of 1 was assigned if present and 0 if absent. Conjunctivochalasis was evaluated at G0, G3, and G7. Figure 5 illustrates conjunctivochalasis under the conditions of G0, G3, and G7.

#### Statistical Analysis

Descriptive statistics were used to describe samples in terms of mean and standard deviation (SD). *p* < 0.05 was considered statistically significant in all analyses. Inter-rater reliability of the above parameters (TBUT, TMH, CFS score, and conjunctivochalasis) was analyzed using intraclass correlation coefficients (ICCs). Values were considered poor (ICC < 0.5), moderate (0.5 ≥ ICC < 0.75), good (0.75 ≥ ICC < 0.90), and excellent (ICC ≥ 0.90) [26]. All analyses were performed using JMP Pro 17 software (SAS Institute, Cary, NC, USA).

## 3. Results

### 3.1. Patients’ Characteristics

This study assessed a total of 92 eyes of 46 subjects. The sample characteristics of this study’s subjects were as follows: 20 males (43.5%) and 26 females (56.5%), with a mean age of 52.22 ± 18.68 years. The mean OSDI score was 22.75 ± 22.03.

### 3.2. InterObserver Reliability

Interobserver reliability statistics are summarized in Table 1. ICC values for TBUT were 0.3046, 0.6124, and 0.7381 for G0, G3, and G7, respectively. For CFS scores, ICC values were 0.4656, 0.8413, and 0.2259 for G0, G3, and G7, respectively. TMH showed ICC values of 0.7221, 0.5774, and 0.6650 for G0, G3, and G7, respectively. Finally, ICC values for conjunctivochalasis were 0.5618, 0.2820, and 0.0786 for G0, G3, and G7, respectively.

## 4. Discussion

The objective of this study was to evaluate how interobserver reliability in the measurement of TBUT and TMH, CFS scoring, and the detection of conjunctivochalasis is affected by different levels of image enhancement: no enhancement (G0), mild enhancement (G3), and strong enhancement (G7). The results demonstrated that as the intensity of image enhancement increased, TBUT became more easily detectable. G7 was considered optimal for the detection of subtle tear film break-up. In contrast, for CFS scoring, mild enhancement (G3) improved the visibility of corneal staining, while excessive enhancement (G7) reduced clarity. Therefore, G3 was deemed most suitable for CFS evaluation. Regarding the detection of conjunctivochalasis, greater image enhancement progressively hindered its identification, making G0 the most appropriate setting. Similarly, higher levels of enhancement impaired the accuracy of TMH measurements, suggesting that G0 is preferable for TMH assessment. These findings underscore the importance of selecting an appropriate level of image enhancement tailored to the specific diagnostic parameter under investigation. Furthermore, this study’s results indicate that with appropriate image enhancement, general ophthalmologists can achieve diagnostic accuracy comparable to that of dry eye specialists.

The SEC is a novel, portable device specifically developed for ophthalmic examinations. This innovative technology converts a standard smartphone camera into a tool capable of performing fundamental assessments of the ocular surface [11]. In addition to its diagnostic utility, although previous reports have indicated a low ICC for TBUT, the SEC has demonstrated outstanding interobserver reproducibility in the evaluation of allergic conjunctivitis and TBUT measurements [14,27]. It is well established that the application of appropriate image enhancement techniques can significantly improve the visualization of target structures during surgeries and in the interpretation of diagnostic images. In ocular surgery, digital technology for three-dimensional (3D) visualization employs real-time filters to enhance contrast and optimize the clarity of anatomical structures [28,29]. Several studies have demonstrated improved visualization using digital imaging systems, particularly in cataract surgery and during the staining of epiretinal membranes in surgical procedures [30,31]. The Olympus Corporation introduced texture and color enhancement imaging in colonoscopy, which enhances subtle tissue textures, selectively brightens dark areas, and accentuates the contrast between red and white colors. These enhancements have been shown to improve the adenoma detection rate [32,33].

In this study, we hypothesized that with appropriate image processing, general ophthalmologists could achieve diagnostic accuracy comparable to that of dry eye specialists when using SEC for anterior segment imaging. To test this, we prepared three versions of each image: the original, unprocessed version (G0), a mildly enhanced version (G3), and a highly enhanced version (G7). We then examined differences in diagnostic outcomes for TBUT, CFS score, TMH, and conjunctivochalasis across these image versions. The findings indicate that, even when diagnosing from the same images, it is essential to select an appropriate level of image enhancement based on the specific structures or conditions being assessed. By applying suitable image processing techniques, general ophthalmologists can achieve diagnostic accuracy comparable to that of dry eye specialists. In addition, this study investigated the potential of image enhancement to achieve diagnostic capabilities comparable to fluorescein staining under a blue filter [21]. By applying software modifications to images captured with the SEC, it is anticipated that the SEC device could eventually facilitate the detection of DED without the need for a blue filter. This advancement may enable non-ophthalmologists and non-physician healthcare providers to effectively diagnose DED using videos captured by the SEC. Such capabilities would render the device especially valuable in regions with limited access to ophthalmologists specializing in DED, thereby addressing unmet needs in DED diagnosis. In this study, images were evaluated on a separate device from the one used for image capture, as the ophthalmologists were located remotely from the imaging site. However, direct diagnosis on the device used for image capture is feasible, which could enhance practicality and efficiency in real-world clinical applications. A recent study comparing ocular surface measurements obtained using three imaging devices—the Keratograph 5M (Oculus, Wetzlar, Germany), the Antares (Lumenis, Sydney, Australia), and the LacryDiag (Quantel Medical, Cournon d’Auvergne, France)—revealed significant variability and moderate reliability among devices [34]. Notably, discrepancies were observed in TMH and NITBUT measurements, depending on the specific device used. To mitigate the challenges posed by subjective variability and the labor-intensive nature of traditional measurement methods, artificial intelligence (AI) is increasingly being utilized [35]. Its application is expected to broaden further in the near future, enhancing precision and efficiency in ocular surface assessment [36,37,38]. To ensure that AI systems provide highly accurate diagnoses, it is important to train them using appropriately processed images tailored to the condition being evaluated.

The limitations of this study include its retrospective design and small sample size. Additionally, the images used in this study were captured using the SEC, which may limit the generalizability of the findings to images obtained from other imaging devices. For the diagnosis of other conditions, such as cataracts, specific image processing parameters would need to be identified for each disease individually. Further studies with larger and more diverse samples would be beneficial to validate these results. In future studies, exploring a yellow filter may be valuable to enhance contrast during imaging.

## 5. Conclusions

Appropriate image processing techniques can significantly enhance the visibility of target diagnostic features, improving diagnostic accuracy. This enables general ophthalmologists to achieve diagnostic accuracy comparable to dry eye specialists. However, as optimal image processing parameters may vary depending on the diagnostic target, it is essential to adjust these parameters accordingly for each condition.

## Figures and Tables

**Figure 1 diagnostics-14-02552-f001:**
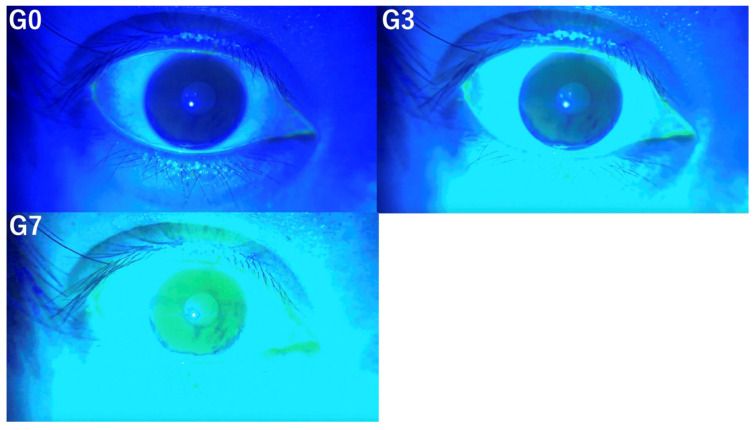
This figure illustrates the same photograph subjected to three different image enhancement techniques, demonstrating how it appears at G0 (**left upper** panel), G3 (**right upper** panel), and G7 (**left lower** panel).

**Figure 2 diagnostics-14-02552-f002:**
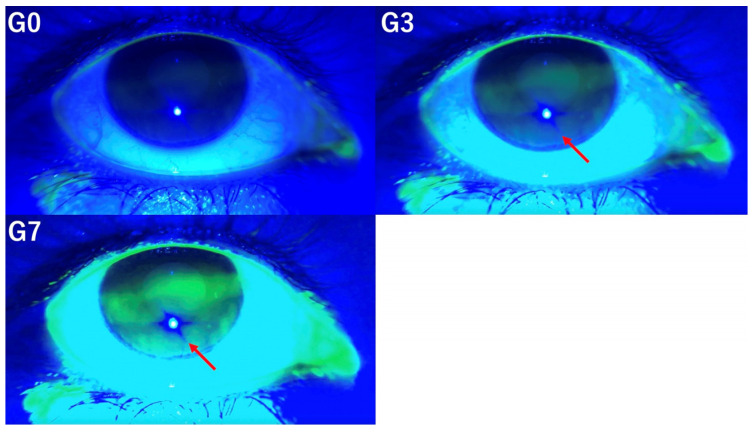
TBUT was evaluated at G0 (**left upper** panel), G3 (**right upper** panel), and G7 (**left lower** panel). These pictures illustrate the appearance of tear film break-up under the conditions of G0, G3, and G7 (arrow).

**Figure 3 diagnostics-14-02552-f003:**
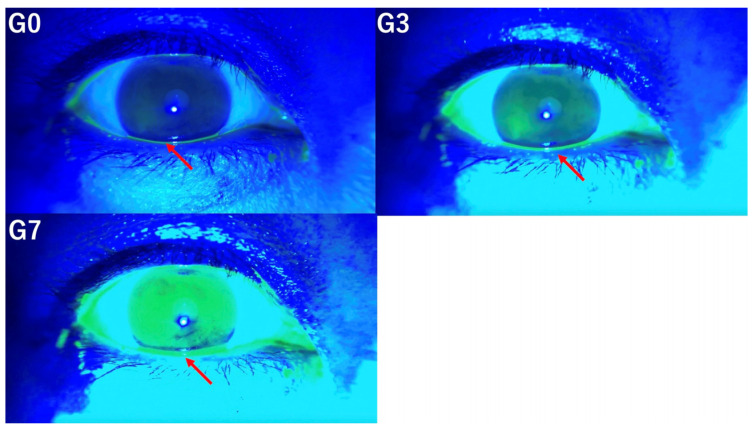
Evaluation of TMH under G0 (**left upper** panel), G3 (**right upper** panel), and G7 (**left lower** panel). These pictures illustrate TMH under the conditions of G0, G3, and G7 (arrow).

**Figure 4 diagnostics-14-02552-f004:**
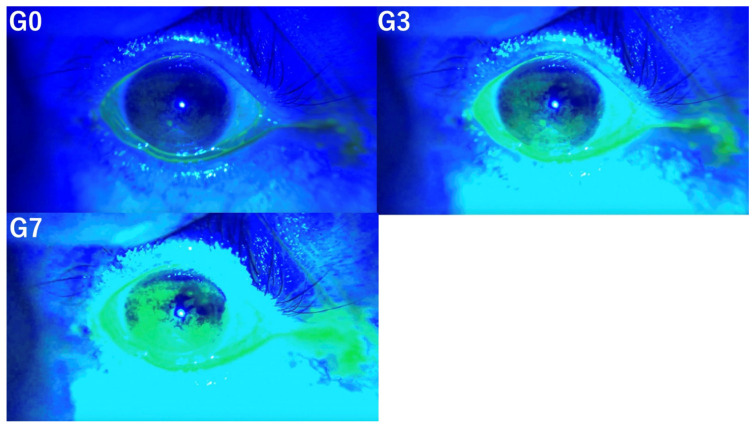
CFS scores were evaluated at G0, G3, and G7. These pictures illustrate corneal fluorescein staining under the conditions of G0, G3, and G7.

**Figure 5 diagnostics-14-02552-f005:**
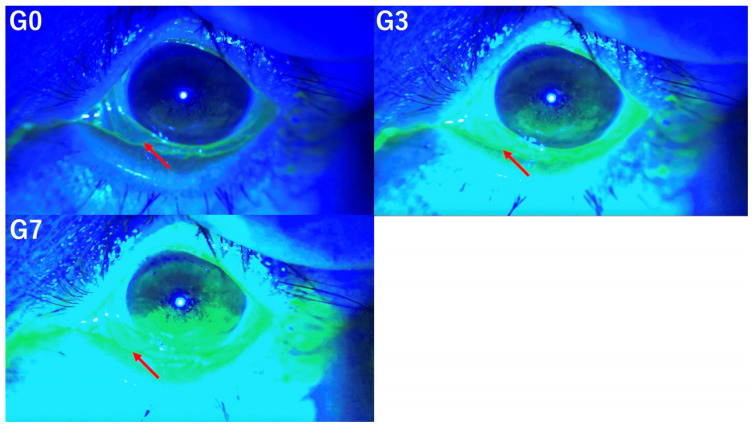
Evaluation of conjunctivochalasis under G0 (**left upper** panel), G3 (**right upper** panel), and G7 (**left lower** panel). These pictures illustrate conjunctivochalasis under the conditions of G0, G3, and G7 (arrow).

**Table 1 diagnostics-14-02552-t001:** Interobserver reliability statistics.

	Image Enhancement	ICC Values
TBUT	G0	0.3046
	G3	0.6124
	G7	0.7381
CFS scores	G0	0.4656
	G3	0.8413
	G7	0.2259
Conjunctivochalasis	G0	0.5618
	G3	0.2820
	G7	0.0786
TMH	G0	0.7221
	G3	0.5774
	G7	0.6650

CFS, corneal fluorescein staining; G0, no enhancement; G3, mild enhancement; G7, high enhancement; ICC, intraclass correlation coefficients; TBUT, tear break-up time; TMH, tear meniscus height.

## Data Availability

Data used and analyzed for this study are available from the corresponding author on reasonable request.

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
