# Peer review of "Evaluating the Effect of Image Enhancement on Diagnostic Reliability in Dry Eye Disease Using a Portable Imaging Device"

_diagnostics, 2024, doi:10.3390/diagnostics14222552_

Round 1

Reviewer 1 Report

Comments and Suggestions for Authors

1.The authors state that they chose a specific age range for patient inclusion, between 18 and 95 years, because older patients would show more senile ocular surface alterations. However, since it is known that these senile surface changes occur with higher prevalence starting from age 60, the rationale for the broader age range is unclear. Please delete this sentence.

2. Figure 2: Please adjust the position of the arrow to clearly indicate the subtle tear film changes. Currently, the arrow is not placed consistently across the images (G7 vs. G0/G3) and points at the corneal limbus rather than at the tear film breakups. Ensure the arrow consistently highlights the tear film changes in each image.

3.
There are several imaging devices available on the market for the DED diagnosis, such as the Keratograph 5M, Antares Plus, and IDRA. It is crucial to include a discussion of these devices, focusing on their specific advantages and limitations in the context of DED diagnosis. Moreover, as the focus of this paper is on diagnostic accuracy and examiner independence, it is necessary to compare the interobserver reliability of each device.

4.
What potential applications does the SEC device with image processing techniques offer? Additionally, what would be the therapeutic implications of diagnosing DED in regions where ophthalmologists specializing in DED are scarce? Including this discussion would provide valuable context for readers, helping them understand the practical applications of the SEC device and its image processing methods.

5. The study shows that different enhancement levels are optimal for diagnosing different DED parameters such as tear break-up time, tear meniscus height, corneal fluorescein staining, and conjunctivochalasis. In this manuscript, the image enhancement was performed on a separate computer, not on the phone used to record the images. Please elaborate on whether it would be possible to perform a simplified evaluation directly on the phone that records the images. Otherwise, the need to transfer and evaluate the images on a computer represents a significant disadvantage of this method, potentially limiting its practicality and efficiency in clinical settings.

Author Response

Dear Reviewer,

We would like to express our sincere gratitude for your thoughtful review of our manuscript. Your insightful comments and constructive feedback have greatly enhanced the quality of our work.

We are particularly appreciative of your specific suggestions, which have allowed us to deepen the content of the manuscript significantly. We believe that the revisions made in response to your feedback have resulted in a much-improved version.

Thank you once again for your valuable contributions.

1.The authors state that they chose a specific age range for patient inclusion, between 18 and 95 years, because older patients would show more senile ocular surface alterations. However, since it is known that these senile surface changes occur with higher prevalence starting from age 60, the rationale for the broader age range is unclear. Please delete this sentence.

Response: Thank you for your valuable comment. We agree with your assessment, and accordingly, we have deleted the sentence as suggested. (Line 109-111)

  1. Figure 2: Please adjust the position of the arrow to clearly indicate the subtle tear film changes. Currently, the arrow is not placed consistently across the images (G7 vs. G0/G3) and points at the corneal limbus rather than at the tear film breakups. Ensure the arrow consistently highlights the tear film changes in each image. 

Response: Thank you for your valuable comment. As you noted, the position of the arrow was inconsistent and did not accurately indicate the tear film changes. We have corrected the arrow placement to consistently highlight the tear film breakup areas in G3, and G7. (Line 222)

  1. There are several imaging devices available on the market for the DED diagnosis, such as the Keratograph 5M, Antares Plus, and IDRA. It is crucial to include a discussion of these devices, focusing on their specific advantages and limitations in the context of DED diagnosis. Moreover, as the focus of this paper is on diagnostic accuracy and examiner independence, it is necessary to compare the interobserver reliability of each device.

Response: Thank you for your valuable feedback regarding the discussion of imaging devices for DED diagnosis. In response to your suggestion, I have added a section addressing the Non-Invasive Break-Up Time (NIBUT) and its relevance to DED diagnosis. I appreciate your guidance in enhancing the manuscript's focus on diagnostic accuracy and interobserver reliability. (Line49-52)

Thank you once again for your insightful comments.

  1. What potential applications does the SEC device with image processing techniques offer? Additionally, what would be the therapeutic implications of diagnosing DED in regions where ophthalmologists specializing in DED are scarce? Including this discussion would provide valuable context for readers, helping them understand the practical applications of the SEC device and its image-processing methods.

Response:

Thank you for your insightful comment regarding the potential applications of the SEC device and its image-processing techniques. We are conducting this research with the prospect that, in the future, image enhancement could enable DED detection under fluorescein staining without the need for a blue filter, achieving comparable diagnostic capability. If successful, this advancement would allow non-ophthalmologists or non-physician healthcare providers to diagnose DED using videos captured with the SEC device. This would make the device a valuable tool in regions where ophthalmologists specializing in DED are scarce, broadening access to diagnosis and timely intervention.  (Line318-328)

  1. The study shows that different enhancement levels are optimal for diagnosing different DED parameters such as tear break-up time, tear meniscus height, corneal fluorescein staining, and conjunctivochalasis. In this manuscript, the image enhancement was performed on a separate computer, not on the phone used to record the images. Please elaborate on whether it would be possible to perform a simplified evaluation directly on the phone that records the images. Otherwise, the need to transfer and evaluate the images on a computer represents a significant disadvantage of this method, potentially limiting its practicality and efficiency in clinical settings.

Response: Thank you for your insightful comment. In this study, three ophthalmologists diagnosed the images on a separate device to allow for assessment in locations different from where the images were captured. However, it is certainly possible to perform the diagnosis directly on the device used to capture the images. (Line328-331)

Reviewer 2 Report

Comments and Suggestions for Authors

In the present study, the authors aim to compare the evaluation of two types of observers—experienced and inexperienced—in assessing a group of different parameters using a portable camera. The rationale behind the study is well explained in the introduction, and the topic is highly interesting, as diagnostic procedures and reliability are crucial in ocular surface exploration. However, the authors may need to revise the entire manuscript. There is a significant lack of references, and both the results and discussion sections require further review. Please find below the detailed comments regarding these issues.

In addition, The general citation of references in the text should be revised to align with the journal's format (numbers should be placed in brackets and separated by a space from the preceding word).

Line 19: I would suggest specifying that each parameter/test has its own set.

Lines 35-37: While this is correct, it might be beneficial to update the description of the risk factors, both environmental and inherent to the participants, to align with more recent literature (PMID: 37062428, PMID: 37100346, PMID: 39074684).

Line 45: It would be helpful to include a reference for this sentence (PMID: 31425351).

Line 83: I would recommend using the past tense here, as the study has already been completed.

Lines 97-98: This sentence would benefit from a reference to support the reliability of the design employed (PMID: 31772113).

Lines 100-107: Additional references may be needed in this section to support the statements made.

Line 129: It would be useful to report the brand and manufacturer of the fluorescein. Additionally, clarifying whether the fluorescein was instilled using a micropipette or directly as a drop would enhance the clarity of the text.

Lines 171-172: Since these acronyms were already mentioned earlier, it might be more concise to use only the acronyms here and omit the full terms.

Lines 173-174: I would suggest adding the criteria used to classify observers as experts or non-experts. This could enhance the reproducibility of the study and help readers better understand the manuscript's scope. This is an important point, as the difference in experience among observers is one of the manuscript's main considerations.

General considerations (Lines 139-169 and images 181-183): Could you please clarify why a yellow filter was not used in the analysis? Based on the images, it seems that only cobalt blue light was used and that the images required post-production enhancement for better visualization of the structures. It seems possible that this lengthy process might be simplified by adding a simple and affordable (as suggested in the comments in Lines 50-55) yellow filter to the camera's illumination. Additionally, there are now cost-effective camera adapters available on the market that include both blue light and a yellow filter (https://www.visuscience.com/en/product/15). Could you please elaborate on the specific advantages of the present method?

Lines 186-187: This classification method requires a reference, as previous scales have been proposed and tested (PMID: 2028774).

Lines 207-241: This section appears quite subjective, with limited reference to actual results. The considerations mentioned seem to be based more on the authors' impressions rather than statistical outcomes, which could potentially weaken the report. This is particularly relevant since the interclass correlation analysis is typically used to verify the reliability of different contrast methods. I would suggest considering the removal of this section and instead presenting the images as examples, without including personal interpretations, perhaps in the methodology section.

Lines 254-256: The acronyms do not need to be defined again at this point in the manuscript and can be used without restating their full meaning. This issue with the acronyms appears several times throughout the discussion section. I would recommend a thorough review of this section to address this matter.

Discussion general comment: The discussion section may need revision. The authors tend to repeat information already presented earlier in the manuscript and focus primarily on their own results. It is crucial in studies of this nature to conduct a thorough literature review, comparing the results with similar studies where, for example, interobserver comparisons of TBUT and TMH evaluations were performed using innovative methods. This comparison helps to determine whether their findings are similar, better, or worse, and to better understand the true impact of the manuscript and its contribution to the scientific community.

Comments on the Quality of English Language

No issues with the English quality

Author Response

Dear Reviewer,

Thank you very much for your thoughtful review of our manuscript. We greatly appreciate the time and effort you invested in providing feedback and for recommending several valuable references. Your insights will significantly enhance the depth and quality of our work.

We are committed to addressing your comments and suggestions, which we believe will lead to a stronger manuscript. Your contributions are invaluable, and we look forward to presenting a revised version that reflects your recommendations.

Thank you once again for your support.

・The general citation of references in the text should be revised to align with the journal's format (numbers should be placed in brackets and separated by a space from the preceding word).

Response: Thank you for bringing this to our attention. This was an oversight on my part, and I have revised the citations to align with the journal's format, placing numbers in brackets and separated by a space from the preceding word.

・Line 19: I would suggest specifying that each parameter/test has its own set.

Response: Thank you for your suggestion. As noted in Lines 184-187, for the same video, we designated the unprocessed images as G0, and the enhanced images as G3 and G7, according to the degree of image enhancement applied.

・Lines 35-37: While this is correct, it might be beneficial to update the description of the risk factors, both environmental and inherent to the participants, to align with more recent literature (PMID: 37062428, PMID: 37100346, PMID: 39074684).

Response: Thank you for your valuable suggestion. Based on the references you provided, we have added a discussion on the impact of digital screen exposure, which has recently been recognized as a significant factor contributing to dry eye, especially in the context of prolonged exposure. (Line41-42)

・Line 45: It would be helpful to include a reference for this sentence (PMID: 31425351).

Response: Thank you for your suggestion. As you indicated, it is indeed appropriate to include the reference you recommended for this section, and we have now added it accordingly. (Line53)

・Line 83: I would recommend using the past tense here, as the study has already been completed.

Response: Thank you for your valuable suggestion. As recommended, I have revised this section to the past tense. (Line90)

・Lines 97-98: This sentence would benefit from a reference to support the reliability of the design employed (PMID: 31772113).

Response: Thank you for your valuable comment. As you pointed out, including a reference would strengthen the reliability of the design, so I have added a recommended citation to support this section. (Line105)

・Lines 100-107: Additional references may be needed in this section to support the statements made.

Response: Thank you for your valuable feedback. I agree that additional references would enhance this section. Therefore, I have included the relevant literature that informed the establishment of the inclusion and exclusion criteria. (Line115)

・Line 129: It would be useful to report the brand and manufacturer of the fluorescein. Additionally, clarifying whether the fluorescein was instilled using a micropipette or directly as a drop would enhance the clarity of the text.

Response: Thank you for your valuable feedback. I have included the brand and manufacturer of the fluorescein in the revised manuscript. Additionally, as previously mentioned in the text, the fluorescein was instilled using a micropipette. This clarification has been retained to enhance the overall clarity of the manuscript. (Line137-18)

・Lines 171-172: Since these acronyms were already mentioned earlier, it might be more concise to use only the acronyms here and omit the full terms.

Response: Thank you for your valuable feedback. I have removed the full terms as suggested and will use only the acronyms in the revised manuscript. (Line179-180)

・Lines 173-174: I would suggest adding the criteria used to classify observers as experts or non-experts. This could enhance the reproducibility of the study and help readers better understand the manuscript's scope. This is an important point, as the difference in experience among observers is one of the manuscript's main considerations.

Response: Thank you for your valuable suggestion. In response, we have clarified the criteria for classifying observers as experts. Observers with over five years of experience in a dedicated dry eye clinic were defined as dry eye specialists. This criterion has now been added to the manuscript to enhance its reproducibility and provide clearer context regarding observer expertise. (Line182-183)

・General considerations (Lines 139-169 and images 181-183): Could you please clarify why a yellow filter was not used in the analysis? Based on the images, it seems that only cobalt blue light was used and that the images required post-production enhancement for better visualization of the structures. It seems possible that this lengthy process might be simplified by adding a simple and affordable (as suggested in the comments in Lines 50-55) yellow filter to the camera's illumination. Additionally, there are now cost-effective camera adapters available on the market that include both blue light and a yellow filter (https://www.visuscience.com/en/product/15). Could you please elaborate on the specific advantages of the present method?

Response: Thank you for this insightful comment. In this study, we intentionally used images without a yellow filter, as our primary aim was to explore image analysis methods that could eventually replace the need for physical filters. By working with unfiltered images, we are developing and optimizing algorithms that may allow for accurate visualization and analysis without the addition of a yellow filter. This approach could simplify image capture and reduce the need for additional equipment, potentially enhancing the accessibility and efficiency of diagnostic imaging in clinical settings.

・Lines 186-187: This classification method requires a reference, as previous scales have been proposed and tested (PMID: 2028774).

Response: Thank you for your valuable feedback. As you pointed out, previous scales have indeed been proposed and tested, and I have added the paper you suggested as a reference to support this classification method. (Line196)

・Lines 207-241: This section appears quite subjective, with limited reference to actual results. The considerations mentioned seem to be based more on the authors' impressions rather than statistical outcomes, which could potentially weaken the report. This is particularly relevant since the interclass correlation analysis is typically used to verify the reliability of different contrast methods. I would suggest considering the removal of this section and instead presenting the images as examples, without including personal interpretations, perhaps in the methodology section.

Response: Thank you for your valuable feedback. As you suggested, we have revised the section to present the images as examples without adding our interpretations. We agree that removing subjective commentary strengthens the results and aligns better with the intended use of interclass correlation analysis to verify the reliability of the contrast methods.

Thank you again for helping us improve the clarity and objectivity of our report. (Line216-261)

・Lines 254-256: The acronyms do not need to be defined again at this point in the manuscript and can be used without restating their full meaning. This issue with the acronyms appears several times throughout the discussion section. I would recommend a thorough review of this section to address this matter.

Response: Thank you for your valuable feedback. I have removed the full terms as suggested and will use only the acronyms in the revised manuscript. (Line275,291)

・Discussion general comment: The discussion section may need revision. The authors tend to repeat information already presented earlier in the manuscript and focus primarily on their own results. It is crucial in studies of this nature to conduct a thorough literature review, comparing the results with similar studies where, for example, interobserver comparisons of TBUT and TMH evaluations were performed using innovative methods. This comparison helps to determine whether their findings are similar, better, or worse, and to better understand the true impact of the manuscript and its contribution to the scientific community.

Response: Thank you very much for your valuable feedback. As you pointed out, we have expanded the discussion section, incorporating comments from the other reviewer as well. Additionally, we included findings from a study using animal models that reported on the ICC of TBUT, allowing for a more comprehensive comparison with our results. (Line294-295)

Thank you again for your insightful suggestions, which have greatly contributed to enhancing the manuscript.

Round 2

Reviewer 2 Report

Comments and Suggestions for Authors

I would like to thank the authors for their dedicated efforts in revising the manuscript and for their thorough responses to the previous comments. The updated version of the manuscript demonstrates improvement across several areas. However, some issues remain unresolved, and I would recommend a further revision to enhance the clarity and rigor of the work. Please find my additional comments and suggestions below.

Line 19: Thank you for the clarification. However, I believe I may not have conveyed my initial feedback clearly. I would recommend specifying in the abstract that a separate set of images was created for each enhancement stage (G0, G3, and G7) and that these sets were not intermixed. This distinction would improve clarity regarding the methodological approach.

Lines 168-177: I still consider that the use of a yellow filter does not introduce a significant issue in protocols of this kind. However, I recommend simplifying the post-production process, and enhancing or standardizing the images in a way that ensures appropriate visualization without overcomplication. It would be beneficial to include this point in the discussion or limitations section to provide a more transparent assessment of this methodological choice.

Line 249-250: Please add a reference for this statement (e.g., PMID: 9829104).

Lines 278-338: I appreciate the authors' revisions to this section. Nonetheless, as mentioned in my previous review, I believe that only Section "3.6. Inter-observer reliability" aligns with the study objectives as outlined in lines 14-17 and 107-111. The remaining content does not report study results but rather methodological details. I recommend relocating these images to the “2.5. Image Enhancement” section, where they would serve better as methodological illustrations.

Lines 368-447: Although the discussion section has been revised, it remains primarily a summary of study results. I encourage the authors to enrich this section by exploring the implications of their findings, considering future research directions, and comparing their work with studies using alternative devices or methodologies for similar parameters (e.g., PMID: 26948001, PMID: 36865061, PMID: 35966863). A more comprehensive literature review in this section would provide valuable context and underscore the relevance of the findings.

Author Response

Dear Reviewer,

Thank you very much for your thoughtful review and constructive comments on our revised manuscript. We appreciate the time and effort you have dedicated to reviewing our work and for providing insightful suggestions that have significantly contributed to enhancing the clarity and rigor of our study.

Incorporating your feedback has indeed helped us refine our methodology and improve the overall quality of our manuscript. We will carefully address each of your comments in our next revision to further deepen the content and present our findings with improved clarity.

Once again, thank you for your invaluable input and support in strengthening our manuscript.

・Line 19: Thank you for the clarification. However, I believe I may not have conveyed my initial feedback clearly. I would recommend specifying in the abstract that a separate set of images was created for each enhancement stage (G0, G3, and G7) and that these sets were not intermixed. This distinction would improve clarity regarding the methodological approach.

Response: Thank you for your insightful feedback. We have revised the abstract to specify that a separate set of images was created for each enhancement stage (G0, G3, and G7) without intermixing between sets. We believe this addition enhances the clarity of our methodological approach, as you recommended. (Line 19-21)

Thank you again for helping us improve our manuscript.

・Lines 168-177: I still consider that the use of a yellow filter does not introduce a significant issue in protocols of this kind. However, I recommend simplifying the post-production process, and enhancing or standardizing the images in a way that ensures appropriate visualization without overcomplication. It would be beneficial to include this point in the discussion or limitations section to provide a more transparent assessment of this methodological choice.

Response: Thank you very much for your valuable suggestion regarding the yellow filter. In response, we have added a statement to the limitations section, noting that the use of a yellow filter may be considered in future studies to achieve appropriate visualization. This addition aims to provide greater transparency in our methodological choices, as you recommended. (Line 453-454)

We appreciate your guidance in improving the clarity of our manuscript.

Line 249-250: Please add a reference for this statement (e.g., PMID: 9829104).

Response: We appreciate your suggestion regarding the addition of a reference for the statement. We have now included the reference you provided (PMID: 9829104) in the revised manuscript. (Line 249) Thank you for your valuable input in enhancing the credibility of our work.

・Lines 278-338: I appreciate the authors' revisions to this section. Nonetheless, as mentioned in my previous review, I believe that only Section "3.6. Inter-observer reliability" aligns with the study objectives as outlined in lines 14-17 and 107-111. The remaining content does not report study results but rather methodological details. I recommend relocating these images to the “2.5. Image Enhancement” section, where they would serve better as methodological illustrations.

Response: Thank you very much for your valuable feedback. Following your suggestion, we have moved the images and the relevant content to Section "2.6. Image Analysis" in the Methods section, where they now serve as methodological illustrations as recommended. (Line 248-279)

We appreciate your guidance in enhancing the clarity and alignment of the manuscript with the study objectives.

・Lines 368-447: Although the discussion section has been revised, it remains primarily a summary of study results. I encourage the authors to enrich this section by exploring the implications of their findings, considering future research directions, and comparing their work with studies using alternative devices or methodologies for similar parameters (e.g., PMID: 26948001, PMID: 36865061, PMID: 35966863). A more comprehensive literature review in this section would provide valuable context and underscore the relevance of the findings.

Response: Thank you for your valuable feedback regarding the discussion section of our manuscript. We appreciate your suggestion to enrich this section by exploring the implications of our findings.

We have taken your recommendations to heart and have added the cited references (PMID: 36865061, PMID: 35966863) to provide a more comprehensive literature review. This addition has allowed us to enhance the discussion and provide valuable context that underscores the relevance of our findings. (Line 437-444)

Thank you once again for your insightful comments, which have significantly contributed to improving our manuscript.

Round 3

Reviewer 2 Report

Comments and Suggestions for Authors

Thank you to the authors for their diligent efforts in revising the manuscript. I have reviewed the latest changes, addressing the primary concerns raised in previous rounds. No further revisions are needed, and I appreciate the work that has been done to improve this submission.